

# Design process and preliminary psychometric study of a video game to detect cognitive impairment in senior adults

Sonia Valladares-Rodriguez[1], Roberto Perez-Rodriguez[1], David Facal[2], Manuel J. Fernandez-Iglesias[1], Luis Anido-Rifon[1] and Marcos Mouriño-Garcia[1]

[1] School of Telecommunication Engineering, University of Vigo, Vigo, Spain
[2] Department of Developmental Psychology, Universidad de Santiago de Compostela, Santiago de Compostela, Spain

Corresponding author
Sonia Valladares-Rodriguez,
soniavr@det.uvigo.es

## ABSTRACT

**Introduction**. Assessment of episodic memory has been traditionally used to evaluate potential cognitive impairments in senior adults. Typically, episodic memory evaluation is based on personal interviews and pen-and-paper tests. This article presents the design, development and a preliminary validation of a novel digital game to assess episodic memory intended to overcome the limitations of traditional methods, such as the cost of its administration, its intrusive character, the lack of early detection capabilities, the lack of ecological validity, the learning effect and the existence of confounding factors.
**Materials and Methods**. Our proposal is based on the gamification of the California Verbal Learning Test (CVLT) and it has been designed to comply with the psychometric characteristics of reliability and validity. Two qualitative focus groups and a first pilot experiment were carried out to validate the proposal.
**Results**. A more ecological, non-intrusive and better administrable tool to perform cognitive assessment was developed. Initial evidence from the focus groups and pilot experiment confirmed the developed game's usability and offered promising results insofar its psychometric validity is concerned. Moreover, the potential of this game for the cognitive classification of senior adults was confirmed, and administration time is dramatically reduced with respect to pen-and-paper tests.
**Limitations**. Additional research is needed to improve the resolution of the game for the identification of specific cognitive impairments, as well as to achieve a complete validation of the psychometric properties of the digital game.
**Conclusion**. Initial evidence show that serious games can be used as an instrument to assess the cognitive status of senior adults, and even to predict the onset of mild cognitive impairments or Alzheimer's disease.

# INTRODUCTION

Schacter and Tulving (*Tulving, 1972*; *Tulving 1983*; *Schacter & Tulving, 1994*) define a memory system in terms of its brain mechanisms, the kind of information it processes, and the principles of its operation. This suggests that memory is the combined total of all mental experiences organized into different systems. These authors claim that memory systems and memory types are two different concepts. In particular, they argue (*Schacter & Tulving, 1994*) that there are five memory systems: namely, episodic memory, semantic memory (both belonging to the declarative or explicit memory type) and procedural memory, primary memory and perceptual system (these last three belonging to the non-declarative or implicit memory type).

Episodic memory is a cognitive system that enables human beings to recall past experiences (*Tulving, 2002*). It is about "what", "where", and "when", that is, about memories that are localizable in time and space (*Lezak, 2004*). The evaluation of episodic memory is of paramount importance in cognitive impairments (e.g., MCI or AD) due to its high discriminative power or diagnostic value (*Albert et al., 2011*). Moreover, it is the most reliable predictor or indicator of the likelihood of conversion of MCI to Alzheimer's disease (*Albert et al., 2011*; *Hodges, 2007*; *Petersen, 2004*; *Morris, 2006*; *Facal, Guàrdia-Olmos & Juncos-Rabadán, 2015*; *Juncos-Rabadán et al., 2012*; *Campos-Magdaleno et al., 2015*). In other words, episodic memory alteration is a criterion for an early diagnosis of AD.

There are two major types of neuropsychological tests to perform a neuropsychological evaluation of episodic memory. On the one hand, we can find tests based on the recalling of stories or texts and, on the other hand, tests based on learning and remembering word lists. We can mention several paper-and-pencil (*Lezak, 2004*) tests based on the second approach, such as the California Learning Verbal Test (CVLT), the Children Memory Scale (CMS), the Rey Auditory Verbal Learning Test (RAVLT), or the Wechsler Memory Scale (WMS-III).

One of the most sophisticated and widely used tests to assess episodic memory is the California Learning Verbal Test (CLVT) (*Strauss, Sherman & Spreen, 2006*). In particular, each trial is based on a list of 16 common words grouped in 4 different semantic categories. There is a second edition, CVLT-II, which has improved normative data, especially regarding sensitivity and specificity (*Delis et al., 2000*). One of the most relevant aspects of CVLT in comparison to other tests targeting episodic memory is that it offers more qualitative information on a large number of useful variables and it also provides normative data (*Lezak, 2004*; *Strauss, Sherman & Spreen, 2006*). These variables are learning rate, strategies of semantic and serial clustering, effects of primacy and reluctance, vulnerability to proactive and retroactive interference, assessment of recognition ability, evocation of memory induced by categories, and frequency of different types of errors (e.g., intrusions, repetitions and false positives).

However, the main drawback of this type of assessment tools—CVLT and other tests mentioned above—is that they do not cover the whole complexity around the performance of episodic memory, which encompasses many more elements than just remembering verbal

information. That is, they do not reflect the real complexity of daily tasks involving this memory system (*Plancher et al., 2012*). In addition, general limitations of paper-and-pencil tests should also be considered, such as the resources required for their administration, their intrusive character (*Chaytor & Schmitter-Edgecombe, 2003*), their delayed detection nature (because seniors often visit the doctor when cognitive impairments are evident), the lack of ecological validity (*Knight & Titov, 2009*) and, finally, the influence of practice or learning effect (*Houghton et al., 2004*) and confounding factors such as age, educational levels or cultural background (*Cordell et al., 2013*).

To overcome these problems, this work introduces a novel approach based on the use of gamification techniques, that is, the use of serious games to evaluate the cognitive status of individuals. This new approach presents some valuable advantages (*Parsons, 2014*) such as the possibility of reproducing environments close to real life by using virtual reality to represent concurrent tasks or dynamic sensory stimuli. It also facilitates the standardization of its administration and makes data collection more efficient, and more specifically the capture of response latency time. Finally, this approach enables the random presentation of stimuli across different trials.

We can find in the literature relevant contributions addressing the study of different cognitive areas through video games and serious games, including episodic memory (*Plancher et al., 2012*; *Sauzéon et al., 2015*; *Jebara et al., 2014*; *Díaz-Orueta et al., 2014*), attention (*Díaz-Orueta et al., 2014*; *Iriarte et al., 2016*; *S et al., 2014*), working memory (*Atkins et al., 2014*; *Hagler, Jimison & Pavel, 2014*) and executive functions (*Werner et al., 2009*; *Nolin et al., 2013*) among others. The works referenced tackle a specific aspect of the broader topic of the introduction of games in neuroscience to detect mental disorders such as Mild Cognitive Impairment (MCI) (*Nolin et al., 2013*; *Raspelli et al., 2011*; *Werner et al., 2009*; *Aalbers et al., 2013*; *Tarnanas et al., 2013*; *Kawahara et al., 2015*; *Fukui et al., 2015*), Alzheimer's Disease (AD) (*Tarnanas et al., 2013*; *Kawahara et al., 2015*; *Fukui et al., 2015*), Traumatic Brain Injury (TBI) (*Wilson et al., 1989*; *Canty et al., 2014*), Chronic Fatigue Syndrome (CFS) (*Attree, Dancey & Pope, 2009*), or Attention Deficit Hyperactivity Disorder (ADHD) (*Parsons et al., 2007*; *Pollak et al., 2009*).

Presently, a major limitation of the introduction of video games for the cognitive assessment is the lack of psychometric and normative studies (*Valladares-Rodríguez et al., 2016*). Both aspects are essential to introduce these new mechanisms in clinical practice; that is, to be considered valid and reliable cognitive assessment tools.

In this paper, a novel video game to assess episodic memory is proposed based on the gamification of CVLT. The aim of this research is to overcome the limitations mentioned above, both from classical tests and from existing video game-based approaches. Namely, the following research question was posed: can a video game be designed and developed to assess episodic memory to predict early cognitive impairments in an ecological and non-intrusive way? Next section on Material and Methods describes the process of design, development and validation of the game. Then, the main outcomes of this research and the evidence obtained on the prediction of MCI using that game is presented from the results obtained from two focus group sessions and a first pilot experiment. Results are

analyzed to assess the degree of achievement of the objectives of this research. Finally, existing limitations are identified and the main contributions of this work are summarized. There is evidence that supports that serious games can be used as an instrument to assess the cognitive status of senior adults, and even to early predict cognitive impairments.

## MATERIALS AND METHODS

A specific methodology for the design of scientific research in the field of Information Systems (IS), namely the DS Research Methodology in IS (*Peffers et al., 2007*), was introduced to support this work. This methodology was validated through several case studies (*Studnicki et al., 1996*; *Rothenberger & Hershauer, 1999*; *Chatterjee et al., 2005*; *Peffers & Tuunanen, 2005*). For example, the CATCH study (*Studnicki et al., 1996*), whose scope is aligned with this research, included the development of a comprehensive validation mechanism for more than 250 medical indicators.

However, some adaptations were made to this methodology to cope with our specific validation needs. For example, a cross-sectional or prevalence study (*Barnett et al., 2012*; *Rosenbaum, 2002*; *Kelsey, 1996*) was required in our case combined with a validation model based on the leave-one-out cross-validation strategy (*Kearns & Ron, 1999*; *Cawley & Talbot, 2003*).

To sum up, a set of activities were identified that enabled the design of an artifact (i.e., a video game) focused on the end user (i.e., senior adults). Then, the video game was developed according to an agile and iterative software development methodology and validated with actual users. A pilot experiment was performed to gather data about the psychometric properties and usability of the video game, and the findings were analyzed using machine learning techniques. Finally, results were communicated to the scientific community by means of this paper.

The sequence of steps above enabled the generation of new knowledge on the working hypothesis to fill the existing research gap. These steps are further discussed below.

### Design of the video game

The video game, named Episodix, was designed according to a methodology based on participatory design focused on the end user. The vision of senior adults captured in participatory design workshops and focus groups (*Diaz-Orueta et al., 2012*; *Liamputtong, 2011*) drove the design process from a technical and aesthetical perspective. Besides, experts in neurology, neuropsychology and geriatrics, and patients' associations (cf. Acknowledgments below) provided the expertise required to ensure the validity of the new tool for cognitive assessment. The collaboration with health professionals (*Brox et al., 2011*) from an early design stage is necessary to guarantee both content and clinical validity.

In relation to the design approach, paper-based tests were taken as a reference trying to be challenging and fun at the same time, with the objective of creating a digital game that has the same construct validity as the original test. Existing recommendations (*Brox et al., 2011*; *Nap et al., 2014*) to develop games attractive to senior players were followed. Then, focus groups would confirm whether the game developed was challenging and fun to play, but also

**Table 1  Structural elements of the Episodix video game.**

| | |
|---|---|
| Game goal or purpose | Cognitive assessment, detection of cognitive impairments through the assessment of episodic memory. |
| Target group | Senior people (+55 years old). |
| Location | A virtual walk around a medium-sized town. |
| Rules | During the walk, everyday life objects are displayed and players have to recall the maximum number of elements possible. |
| Challenge | To recall the maximum number of elements actually displayed, trying to avoid objects in the interference lists. |
| Feedback | Each time a correct object is selected, a star is displayed on top of it to represent a positive score or bonus point. |
| Engagement | Objects and furniture try to reproduce a realistic and personalized virtual walk, thanks to the dynamic displaying of different types of elements common in an urban environment. |

if it had a purpose or value, providing a perceived benefit and, eventually, was acceptable and usable by senior adults. Moreover, we tried to replicate real-life situations using virtual reality to reach an ecological solution that reflects the real complexity of daily tasks involving episodic memory. Table 1 summarizes the main structural elements of the game developed.

As mentioned above, the digital game implemented is based on the gamification of CVLT. In CVLT, a list of words is presented as a shopping list for Monday, and the subject taking the test has to remember as many elements in the shopping list as possible. Then, a second interference list is presented as the Tuesday shopping list. After the recall phase, a third list is produced including items from Monday and Tuesday together with new items, and the subject has to recognize elements from the Monday list (*Delis et al., 1987*).

In turn, the game developed consists on a virtual walk through a medium-sized town (cf. Fig. A1 for a street map of the game). During the walk, objects are presented to the participants—in visual and audio format—integrated naturally in the environment. These objects belong to three different collections of objects, namely List-A or main learning list; List-B or interference list; and List-C or yes/no recognition list, which have an average frequency equal to the original pen-and-paper test. In the same way as the conventional test, Episodix consists of the same set of phases, that is, "Learning" or stimuli presentation and "Recall" or remembering of objects. Players have to learn and recall as many objects as possible across several phases.

Given the importance of the concept of semantic category in CVLT, the translation of these categories to the Episodix game is further discussed below. In the pen-and-paper test, the learning lists A and B consist of four semantic categories, two of them being shared among lists (i.e., fruits and herbs). In the case of Episodix, the same happens, only that more ecological categories were chosen to be integrated in an urban-like walk. In Episodix, List A categories are urban furniture, vehicles, professions and shops; while categories in List B are urban furniture, vehicles, buildings, and tools (i.e., urban furniture and vehicles are shared). Most importantly, the average frequency of each list is the same in the pen-and-paper test and the video game. For this, word frequencies where checked taking as

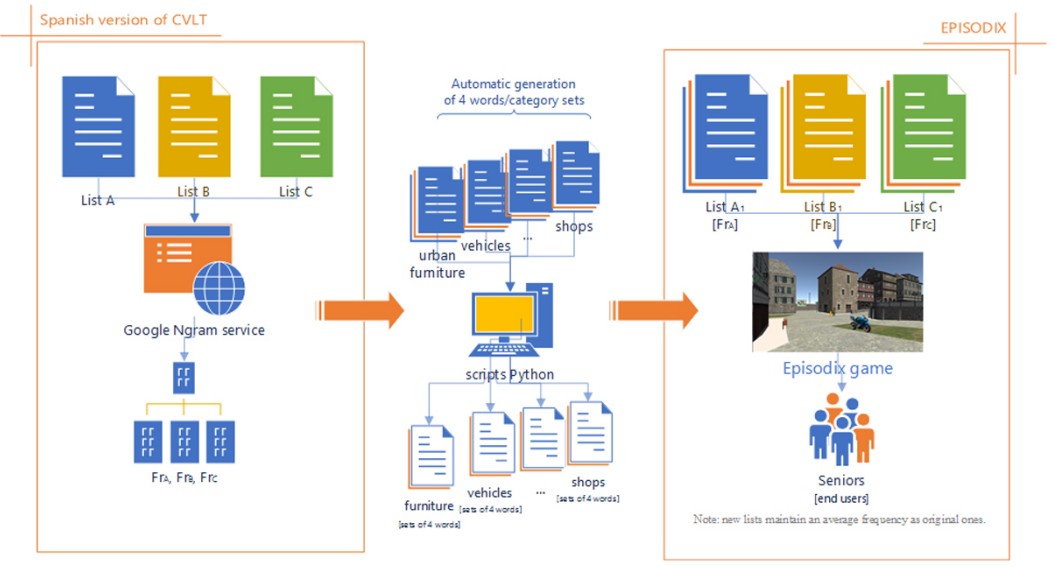

**Figure 1** Process to obtain new object/word list in Episodix game.

a reference a Spanish dictionary of word frequencies (cf. "Diccionario de frecuencias de la unidades lingüísticas del castellano" *Alameda & Cuetos, 1995*) and the corpus offered by the Google n-Gram service (Google books Ngram Viewer: https://books.google.com/ngrams). Next, sets of words were fetched belonging to the new categories having the same average usage frequency as the original test. For this purpose, a computer program was developed to automatically produce sets of four words per category maintaining an average frequency similar to those in the original test. This process is described graphically in Fig. 1. Note that this approach allows us to obtain, in a dynamic and automated way, different versions of the lists to avoid the practice effect across successive administrations.

Furthermore, the temporal administration pattern of Episodix is similar to the original pen-and-paper test (i.e., CVLT). The main difference corresponds to the number of trials and the intermediate break's duration in order to force interferences during subject's recall. Regarding the number of trials, as summarized in Table 2, the administration of List-A is repeated three times, while List-B and List-C, are administered only once. As the game offers a visual reinforcement, it is not necessary to repeat five times the Learning Phase of the first word list. The impact of this approach in the evaluation of the subjects' learning ability is further discussed below (cf. 'Results and Discussion' sections). In relation to the waiting time between short and long term recall, in the original test it is recommended to wait for twenty minutes. This resting time is typically used to administer other neuropsychological tests. In the case of Episodix, it will be suggested to propose the subject to play two or three small games that evaluate other cognitive areas (e.g., attention, executive functions, gnosias, etc.), to guarantee enough waiting time to evaluate long-term recall. These areas, and more specifically attention and executive functions, may serve as early cognitive markers (*Facal, Guàrdia-Olmos & Juncos-Rabadán, 2015*; *Juncos-Rabadán et al., 2012*) for the detection of MCI or initial AD. Finally, apart from the actual administration time,

**Table 2  Pen-and-paper test vs. Episodix video game: administration protocols.**

| Word list | CVLT (pen-and-paper test) | Word list | EPISODIX (video game) |
|---|---|---|---|
| A | Immediate recall (5 trials) | A | Immediate recall (3 trials) |
| B | Immediate recall (1 trials) | B | Immediate recall (1 trials) |
| A | Short-term recall with semantic clues | A | Short-term recall with semantic clues |
| A | Immediate recall (5 trials) | A | Immediate recall (3 trials) |
| | 20′ pause (other tasks carried out) | | 15′ pause (other games played: attention, executive functions, or visual gnosias) |
| A | Long-term recall | A | Long-term recall |
| A | Long-term recall with semantic clues | A | Long-term recall with semantic clues |
| C | Yes/no recognition | C | Yes/no recognition |
| | Manual processing and calculation of assessment results (20′–30′)[a] | | Automatic processing and calculation of assessment results |
| | Expected administration + processing time: 65′–75′ | | Expected administration + processing time: 30′ |

**Notes.**
[a] Apart from the actual administration time, pen-and-paper tests require an additional processing time that depends, among other aspects, on the previous experience of the person administering the test.

pen-and-paper tests require an additional processing time that depends, among other aspects, on the previous experience of the person administering the test. In the case of the video game, this processing is transparent to both the subject under cognitive assessment and the administrator.

Another important design aspect is related to variables or data to be measured in the game. Three categories can be defined according to the granularity or level of detail offered: (i) high level, such as final game scores; (ii) medium level, such as the total number of movements or completed levels; and finally (iii) low level of detail, such as individual responses, accuracy, speed, right actions, errors and omissions. Eventually, low level variables were assessed to be the most appropriate for cognitive evaluation. Moreover, cognitive variables (e.g., yes/no recognition, free recall, short/long delayed recall, recency, primacy, semantic clustering, response to inhibitions, etc.) were also considered in a similar way to the pen-and-paper test (cf. list of variables in Appendix B, Table A1). The analysis of these data will eventually facilitate an assessment of episodic memory, and therefore to obtain some indication on the existence of early cognitive impairments (e.g., MCI or AD).

## Development and implementation of the video game

Concerning the technical development of the video game, an agile and iterative methodology was applied (*Singh, 2008*). This methodology is widely used for the implementation of efficient software solutions. In relation to the development platform, we have chosen Unity (*Lv et al., 2013*), which has a reputable background in the development of 3D games to be run on multiple systems and devices. At the time of writing this paper the sixth development iteration (Third iteration available: http://episodix.gist.det.uvigo.es/.) was completed. The video game is being designed to run on several computer platforms (i.e., Windows, Linux, iOS and Android) and touch devices. It also supports accessible control interfaces such as joysticks, gaming mice, or gesture-controlled devices. Furthermore, Episodix has multilingual design and at the time of writing this paper it supports Spanish, Galician and English. The speed and rate of presented stimuli is adapted to each user

according to their profile, and playing instructions and indicationsare provided in audio and text formats. To sum up, both design and implementation are aimed to facilitate usability and accessibility for any target user, and more specifically for senior adults.

On the other hand, to avoid the learning effect across administrations, Episodix supports multiple scenarios and different collections of objects. A fresh collection is displayed in each game session: List A, B or recognition.

## Validation of the video game

The validation of Episodix is instrumental to guarantee the validity and reliability of the cognitive assessment performed. It involves evaluating the psychometric properties of the video game, which in turn are essential to achieve clinical validity. As a reference for the validation process, we relied on the Standards for Educational and Psychological Testing (*AERA, APA & NCME, 2014*), which state two key attributes for any test: reliability and validity. Reliability indicates the consistency of the test; that is, it should give the same results consistently, as it is supposed to measure the same construct. In turn, validity refers to how well a test measures what it is intended to measure, and includes content validity, criterion validity, face validity, ecological validity, external validity, and construct validity.

Several exploratory experiments were already conducted to evaluate the first four aspects of validity enumerated above (cf. 'Outcomes from focus group sessions'):

- Content validity—Episodix's content is based on the gamification of CVLT, which has been already validated to study episodic memory. Experts in neuropsychological evaluation participating in the design of the game and focus group sessions were instrumental to assess how episodic memory will be evaluated with the game by means of the tasks and variables in it to carry out this evaluation.
- Criterion validity—In our case, the aim was to assess whether or not subjects can be correctly classified only from their interaction with Episodix. For this, data collected from a first pilot experiment was processed using statistical techniques to classify participants in healthy individuals or MCI/AD impaired. Furthermore, we have conducted a correlation study between Episodix's variables and classical tests administered, in order to prove convergent validity a subtype of criterion validity.
- Face validity—In this case, the objective was to check if the game actually appears to measure episodic memory. Meetings with experts in neuropsychological evaluation and the focus group sessions introduced above were used to assess face validity. During these meetings, different solutions for the look and feel of the game were proposed claiming that in all cases the final aim was to evaluate the participants' memory capabilities.
- Ecological validity—Indicates what extent an evaluation instrument approximates real world conditions. Video games based on virtual reality are a reliable instrument to efficiently replicate situations from everyday life. In our case, focus groups were used to gather perceptions about the real-world appearance of Episodix. Moreover, participants in the pilot experiment discussed below were also questioned about this matter among other aspects such as the degree of usability and engagement.

To sum up, these experiences (two focus group sessions and a first pilot experiment), together with several meetings with experts in neuropsychological evaluation, dramatically facilitated the validation of the design of the video game and a preliminary psychometric study of this proposal. The study procedure was approved by the Galician Ethics Committee (Spain) (i.e., code 2016/236).

## Data analysis

Data collected from interactions with Episodix was analyzed using machine learning techniques. Each observation is made up of generic and static data (e.g., age, gender, educational level, socialization level, physical exercise level, frequency of use of technologies, frequency of use of video games, etc.) and the variables obtained in a complete game for each participant. Analysis focused on predictive validity (i.e., predicting performance or behavior from collected data in order to detect cognitive impairment). For that purpose, support vector machine, linear regression and random forest techniques were applied (*Maroco et al., 2011*) as they are widely used in the biomedical field.

For each algorithm an 80/20 data split for training and testing was performed (*Fan et al., 2008*), and the experiment was repeated 5 and 100 times. A higher number of repetitions facilitated the obtaining of stable results, and therefore more representative. Then, the one-left-out (*Soyka et al., 2012*) approach was applied for training/testing each algorithm is trained with the whole sample except one, and then the same algorithm is used to predict the classification of the one left out; and this is repeated as many times as the number of samples available. In this case, the algorithm may be applied to all variables or to a particular subset including the most informative ones. For this, we applied random forest (*Archer & Kimes, 2008*), which facilitates the automatic selection of the most informative variables.

In order to obtain an accuracy value for each method, typical quality metrics about prediction ability were computed (*Sebastiani, 2002*); namely, precision (the ratio of relevant information over total information obtained during the prediction process); F1 (which offers an indication of the significance of predicted and recovered data after classification); sensitivity (the game's ability to identify actually impaired people to avoid false negatives); and specificity (the game's ability to identify healthy people to avoid false positives).

$P\,(Precison) = TP - (TP + FP)$

$F1\,(F1\ score) = 2 * P * R - P + R$

$Sensitivity\,(Recall) = TP - (TP + FN)$

$Specificity = TN - (TN + FP).$

Finally, during the pilot experiment a set of classical tests was administered to participating subjects (i.e., CVLT—Spanish version of California Verbal Learning Test (*Delis et al., 1987*), MMSE—Mini Mental State Examination (*Cockrell & Folstein, 1987*); MAT—Memory Alteration Test (*Rami et al., 2010*); and IQCODE—Informant Questionnaire on Cognitive Decline in the Elderly (*Jorm, 2004*)), in order to gather golden standard data to correlate with Episodix's variables. Classical testing also served to
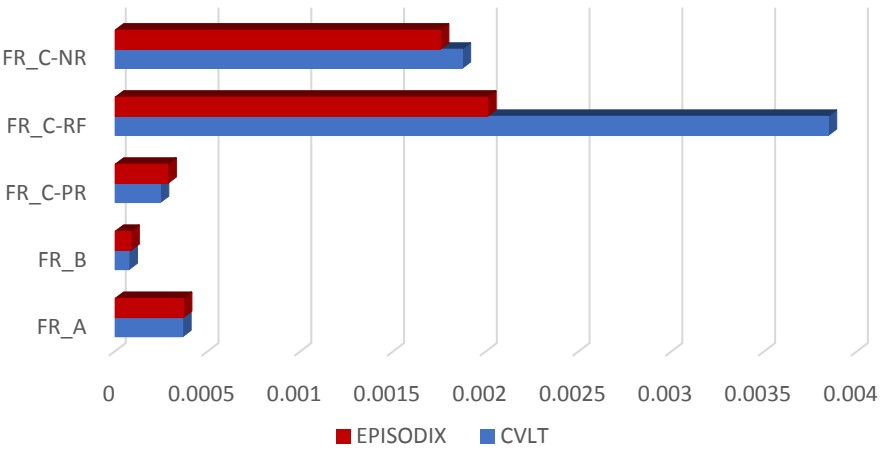

**Figure 2 Frequency comparison of words/objects in CVLT and Episodix.** The vertical axis represents frequency values. vertical axis: FR_A: frequency of List A; FR_B: frequency of List B; FR_C-PR: frequency of prototype word in List C; FR_C-RF: frequency of related word in List C; and FR_C-NR: frequency of nonrelated word in list C.

guarantee that participants were correctly classified according to standard clinical practice, especially the MCI cases. CVLT and MAT were not administered to AD patients to avoid frustration. Moreover, to validate the convergent validity we have conducted a correlation study between Episodix's data and those classical tests. Particularly with CVLT, we can show how the scores in the game world were benchmarked with the results of such cited pen-and-paper test.

## RESULTS

### Outcomes from the selection of the new collection of objects

In order to carry out a first pilot experiment with Episodix, we elaborated three collection of objects equivalent to the original pen-and-paper test in terms of the average frequency of use (cf. Fig. 2), according to the procedure described in 'Design of the video game', Eventually, we compiled a fully functional prototype with the following collections (cf. Tables A2 and A3 in Appendix C):

- List A—composed of 16 objects including urban furniture, vehicles, professions and shops. Their average frequency[1] is $fr_{A-Episodix} = 0.000368$, ($fr_{A-CV\ LT} = 0.000364$ in the case of CVLT).
- List B—16 objects belonging to categories urban furniture, vehicles, buildings, and tools. This collection has an average frequency of $fr_{B-Episodix} = 0.0000845$, in comparison to $fr_{B-CV\ LT} = 0.0000745$ in the case of CVLT.
- List C or recognition—it was constructed following the same criteria as in CVLT, that is, all objects (16) in List A; 8 objects from List B—2 for each common category, and 2 objects from the rest of the categories; 4 prototype objects from List A; 8 objects whose names had a phonetic relationship with elements in List A; and finally, 8 without any phonetic relationship. In this case, frequencies are $fr_{C-Episodix} = 0.000855$ vs. $fr_{C-CV\ LT} = 0.00120$.

[1] It was calculated thorough Google books Ngram Viewer: https://books.google.com/ngrams

To determine the presentation sequence of the objects in each list along the Episodix's virtual city walk, a matching was defined between categories in Episodix and CVLT. Specifically, the matching for List A associates tools with vehicles; herbs with shops; fruits with professions; and clothes with urban furniture. In the case of List B, cooking utensils was bound to vehicles; fruits to buildings; fishes to urban furniture and herbs to tools.

## Outcomes from focus group sessions

The initial goal of focus groups was to gather seniors' opinion on the use of video games in cognitive evaluation in general, and particularly of the digital game developed in this research. The two focus group sessions took place in March–April 2016 (cf. Fig. 3). The first group involved participants in the lifelong learning program of the University of Vigo[2] (three females and two males involved) and the second group was conducted in the premises of the Association of Family Members of Alzheimer Patients and other Dementias of Galicia-Spain (AFAGA in Spanish; http://afaga.com/en/) (six female and one male participants). All participants met the requirements of being +55 years old and active individuals. A high technological or educational level was not required. However, participants had to declare that they did not experience technological rejection or phobia. All participants read the provided patient information sheet and signed informed consent before participation in the sessions. In both documents, consent was requested to record audio speech and take photographs.

Both sessions had a similar duration of around two hours. Participants were offered coffee and pastries in order to get a comfortable and relaxed atmosphere to facilitate proactive participation. The protocol followed during both sessions (cf. Appendix D, Table A4) consisted in an initial introduction of all participants; a brief intervention expressing their opinion about video games and information technologies; a discussion on their understanding of cognitive assessment; a presentation describing the administration of CVLT; a first contact with a very early version of Episodix; and a debate on classical testing vs. the digital game. Finally, they were asked about the perceived usefulness of utilizing games to carry out cognitive evaluation. The outcomes of these sessions included audio files, photographic material and the informed consents signed by all participants. Text transcriptions were obtained from audio files.

The main results are summarized below:

- Content validity—Participants appreciated the idea of the city walk, the new semantic categories and the visual nature of the game rather than a person verbally producing word lists to remember.
- Face validity—Participants perceived the digital game as a useful tool. They described the idea of moving from a pen-and-paper test to a gaming format as interesting and original. However, they would not use it at home because they fear strong concerns about a possible negative outcome. They claimed that it should be a tool to be used by experts in clinical settings. Besides, they seemed motivated and would prefer to play the game instead of taking the pen-and-paper test in the case both instruments were eventually demonstrated to perform the same cognitive assessment. Incidentally, no

[2]Senior program from University of Vigo: https://www.uvigo.gal/uvigo_en/estudos/maiores/index.html.

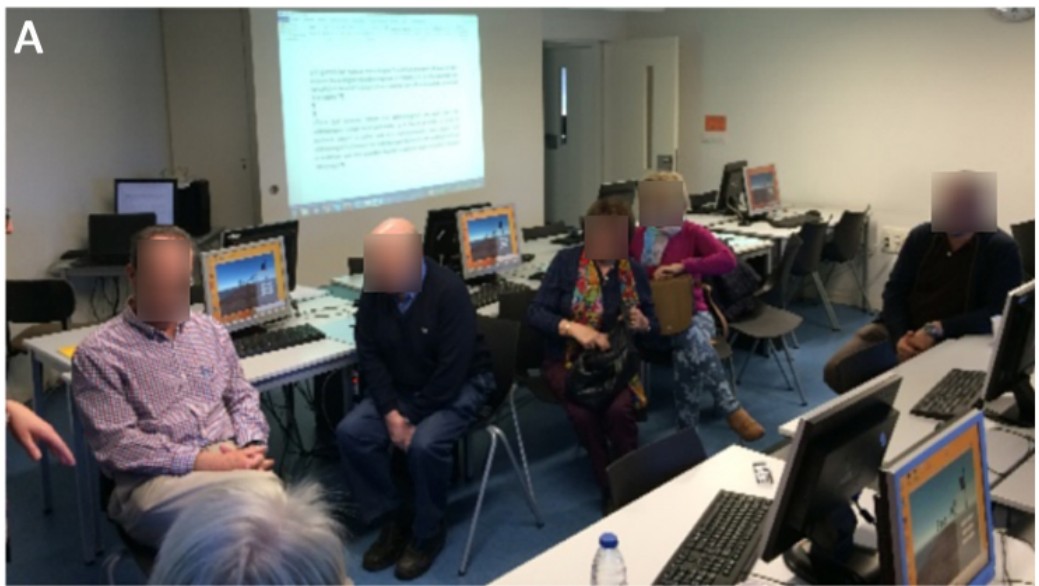

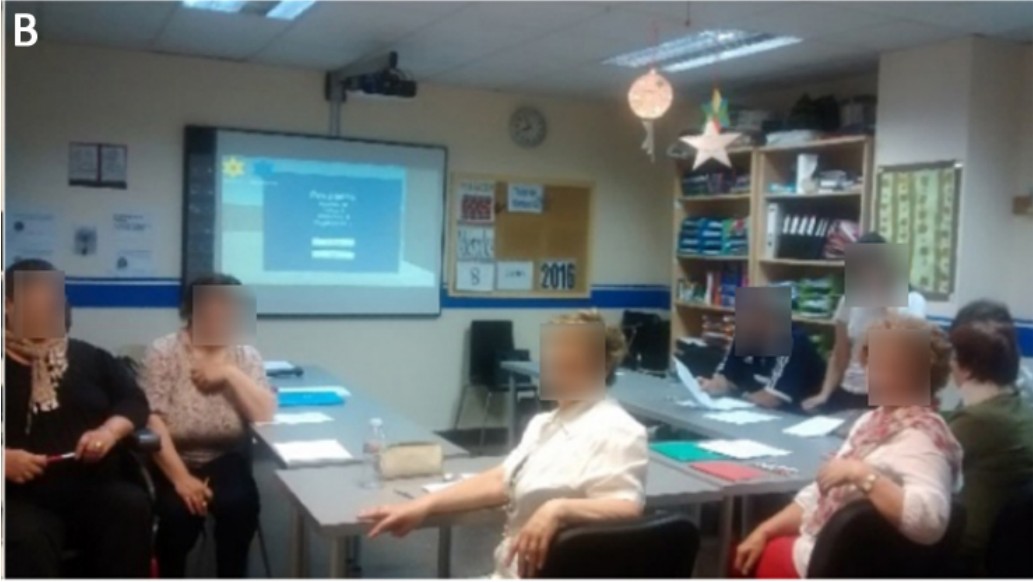

**Figure 3   Focus group sessions with senior people (A at University of Vigo and B at AFAGA).** Photo permission was required and granted in the patient informant sheet. Photo credit: Sonia Valladares-Rodriguez.

matter the game presented was not a digital version of the original test but an actual game, they would prefer the term test to refer to it, as the term video game was perceived to have negative and non-serious connotations.

- Ecological validity—Focus group participants considered both the digital tool and its objects close to real life. In addition, and differently to the pen-and-paper test, the new tool was not perceived as being intrusive.
- Usability: the initial version of Episodix shown was controlled using keyboard and mouse. Participants found it easily administrable, even to use it on their own. However,

they raised some concerns about the interfacing mechanisms, as they claimed that the gaming experience was influenced by users' technological and physical abilities.

## Outcomes from the initial pilot study

The pilot experiment involved a total of 16 subjects, 12 women and four men, living in the southwest area of Galicia (Spain) during May and June 2016. The sample was selected according to a cross-sectional approach, and was divided into three groups: (1) eight people without cognitive impairments or healthy control group (average age of 68.3 ± 8.88 years); (2) five AD patients (average age of 75.8 ± 5.36 years); and finally, three MCI patients (average age of 75 ± 6.08 years). In particular, 12 participants were provided by AFAGA 5 diagnosed with AD, two diagnosed with MCI and five healthy subjects (HC). The four remaining subjects (one MCI patient and three healthy subjects) belonging to the social circle of the authors, were selected randomly within the same geographic area. As in focus group sessions, all participants signed informed consent before participation in this study. The study procedure was approved by the Galician Ethics Committee (Spain) (i.e., code 2016/236). The pilot was conducted at participants' homes (cf. Fig. 4) with the aim of facilitating a relaxing, non-intrusive and close to real-life environment. In other words, cognitive evaluation was carried out in the most ecological and natural environment possible. For that purpose, a more advanced prototype was used than the version introduced during the focus group sessions, more ecological and fully functional.

The pilot experiment consisted of two distinct parts. First, as we have already indicated, a set of classical tests was administered to participating subjects, in order to gather golden standard data to correlate with game, and also, to guarantee that participants were correctly classified (i.e., HC, MCI or AD group). Again, CVLT and MAT were not administered to AD patients to avoid frustration, as suggested by AFAGA.

Participants played a series of games following the time schedule of the original test: the first part of Episodix (to assess immediate and short-term recall), two short-games that evaluate other cognitive areas different from episodic memory; and finally, the second part of Episodix (to gather data about long-delayed recall and recognition ability). Moreover, an initial and final survey was also carried out to capture the user's perception on cognitive games and its possible evolution after playing the digital game.

### *General and cognitive characteristics of subjects*

Participants were characterized according to the parameters below (characteristics are expressed as median values and their standard deviation, due the reduced size of the sample, cf. Fig. 5).

- Frequency of use of ICTs (1[nothing] to 5[a lot]): 3 ± 1.58 for HC subjects; 3 ± 1.53 for MCI subjects and for 1 ± .0 AD patients.
- Frequency of use video games (1[nothing] to 5[a lot]): 1 ± .74 for HC subjects; 3 ± .58 for MCI subjects and for 1 ± .55 AD patients.
- Educational level. 25% of HC subjects completed university studies, 37.5% primary studies, and only 12% have not attended school. In the case of MCI subjects, one of them
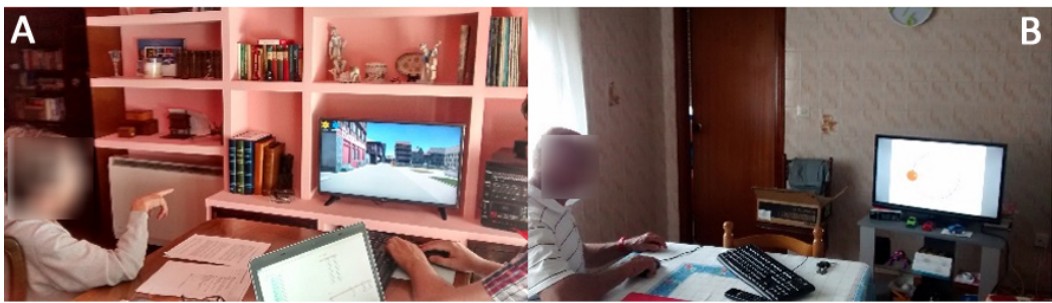

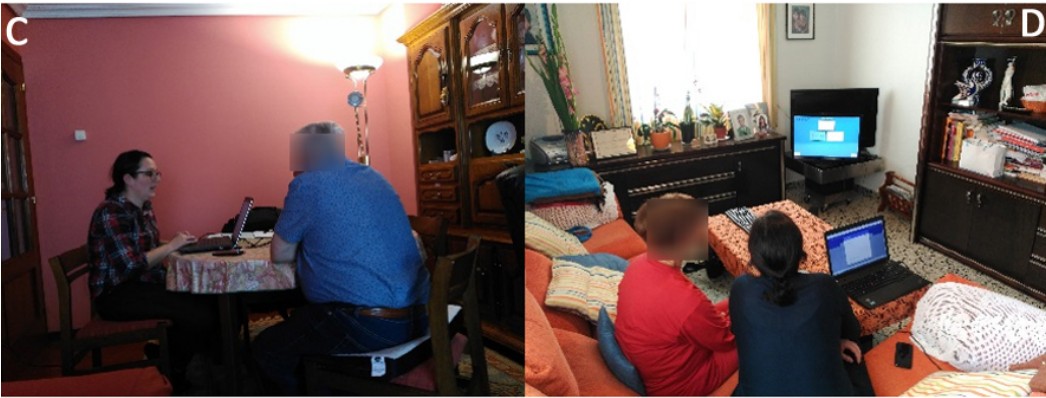

**Figure 4** **Some participants in the pilot (A–D).** Photo permission was required and granted in the patient informant sheet. Photo credit: Sonia Valladares-Rodriguez and Carlos Costa-Rivas.

had university studies, another one attended primary school, and finally one had only reading and writing abilities. AD patients had the lowest educational level: 80% primary and 20% only reading/writing abilities.

- Socialization level (1[being alone most of the time] to 5[socializing a lot): 4.5 ± .74 for HC subjects; 3 ± 1.00 for MCI subjects and for 3 ± .89 AD patients.
- Physical exercise (1[nothing] to 5[a lot]): 3.50 ± 1.41 for HC subjects; 3.50 ± 1.00 for MCI subjects and for 2 ± 1.09 AD patients.
- Classical test: results are summarized in the three graphics at the bottom of Fig. 5.

Blue box plots in Fig. 5 are distributed by cognitive groups, while red dashes indicate median values.

### Usability study

Since it was the first experiment of using Episodix in a real-world situation, we paid close attention to usability and motivational aspects. Usability analysis was based on a 5-point Likert scale questionnaire to facilitate statistical analysis, ranging from strongly disagree (1 point) to strongly agree (5 points). The questionnaire is based on the TAM—Technology Acceptance Model (*Lee, Kozar & Larsen, 2003*), which is considered more flexible to be adapted to different population samples in general, and to senior adults (*Costa et al., 2017*) in particular. TAM indicates that the attitude towards a given technological system is determined by two subjective variables; namely, perceived ease of use (which provides an

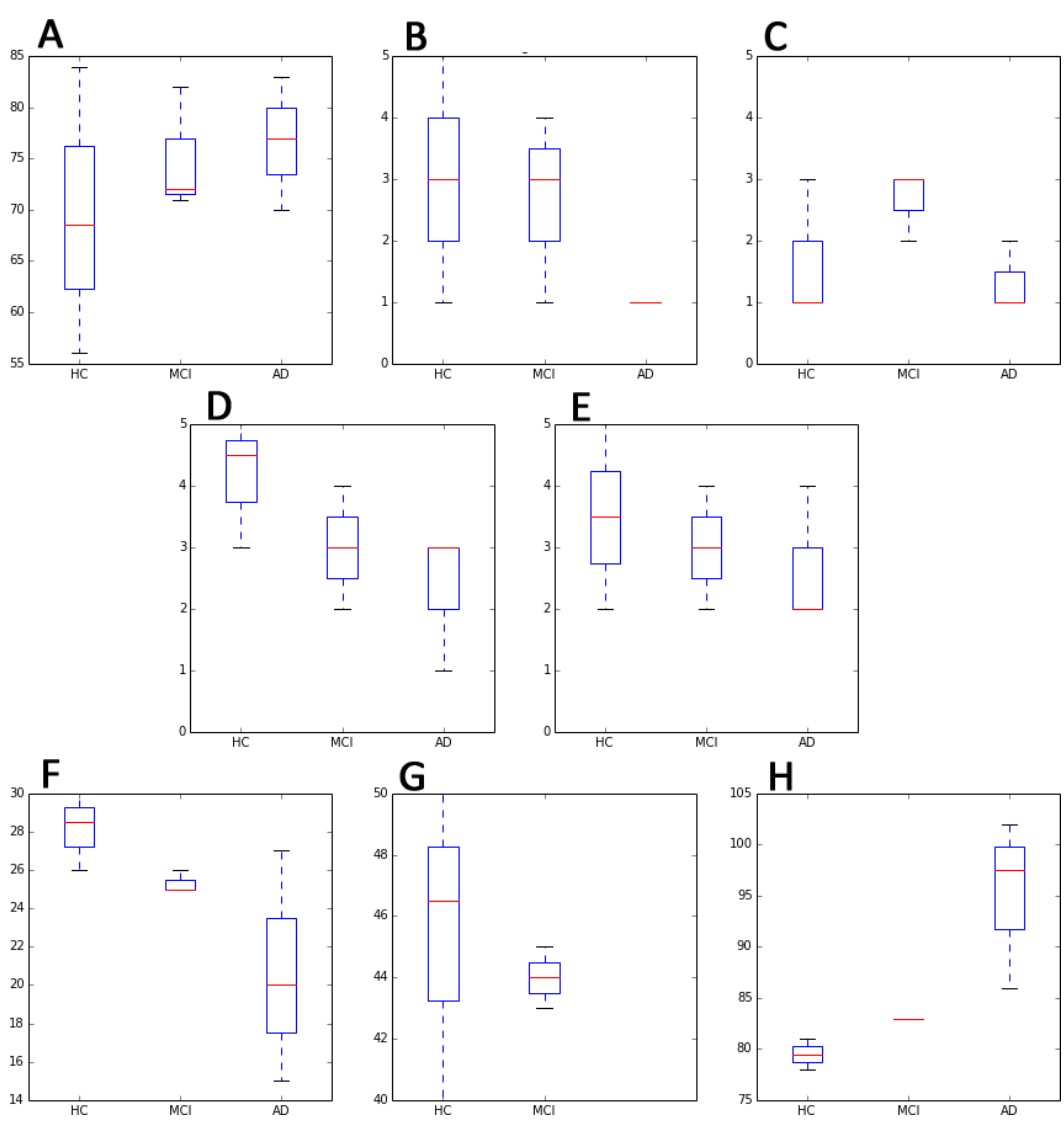

**Figure 5** **Main test subjects' characteristics, distributed by cognitive groups (i.e., HC, MCI and AD).** In red is expressed the median value. (A) AGE; (B) Usage fz. ICTs; (C) Usage fz. Games; (D) Socialize Level; (E) Physical Exercise level; (F) MMSE [score]; (G) MAT [score] and (H) IQCODE [score]. Blue box plots are distributed by cognitive groups.

indication about the extent that a given individual believes that a particular technological system is easy to use) and perceived usefulness (which refers to the extent that a given individual believes that, using a particular technological system, his or her performance would improve). The questionnaire helped to capture the evolution of these aspects before taking part in the pilot and after using the game (cf. Table 3).

Initially, participants stated that they played very little with video games, actual values being 1 for HC people; 2 for MCI subjects; and a 1 for AD seniors (median values).

Valladares-Rodriguez et al. (2017), *PeerJ*, DOI 10.7717/peerj.3508

**Table 3  Comparative motivational and usefulness perceptions of pre-pilot and post-pilot surveys (5-point Likert scale: 1[strongly disagree] to 5[strongly agree]).**

| | HC | MCI | AD | TOTAL | HC rate increase (%) $t$-student | MCI rate increase (%) $t$-student | AD rate increase (%) $t$-student | TOTAL rate increase (%) $t$-student |
|---|---|---|---|---|---|---|---|---|
| [pre] to play VG | 1 | 2 | 1 | 1 | Rate = 40% | Rate = 20% | Rate = 40% | Rate = 40% |
| [post] to play more VG | 3 | 3 | 3 | 3 | $t = 0.005$ | $t = 0.183$ | $t = 0.252$ | $t = 0.0004$ |
| [pre] motivation to play VG | 1 | 3 | 2 | 1 | Rate = 50% | Rate = 0% | Rate = 20% | Rate = 40% |
| [post] more motivation to play VG | 3.5 | 3 | 3 | 3 | $t = 0.021$ | $t = 0.215$ | $t = 0.092$ | $t = 0.001$ |
| [pre] useful CVG | 3 | 4 | 4 | 3.5 | Rate = 30% | Rate = 0% | Rate = −10% | Rate = 10% |
| [post] useful CVG | 4.5 | 4 | 3.5 | 4 | $t = 0.028$ | $t = 0.319$ | $t = 1$ | $t = 0.041$ |
| [pre] useful CVG for you | 2.5 | 4 | 4 | 3 | Rate = 20% | Rate = 0% | Rate = −10 | Rate = 20% |
| [post] useful CVG for you | 3.5 | 4 | 3.5 | 4 | $t = 0.155$ | $t = 0.495$ | $t = 0.495$ | $t = 0.249$ |
| [pre] to use ICTs | 3 | 3 | 1 | 1.5 | Rate = 10% | Rate = 20% | Rate = 20% | Rate = 40% |
| [post] to use more ICTs | 3.5 | 4 | 2 | 3.5 | $t = 0.654$ | $t = 0.391$ | $t = 0.130$ | $t = 0.151$ |
| [post] VG easier than tests | 5 | 4 | 3.5 | 4 | | | | |
| [post] VG more engaging than tests | 3.5 | 4 | 4 | 4 | | | | |
| [post] VG more ecological than tests | 5 | 4 | 3.5 | 4 | | | | |
| [post] VG less intrusive than tests | 4 | 3 | 3 | 4 | | | | |

**Notes.**

VG, video game; CVG, video game for cognitive assessment.

Rows [2:5] express median value by cognitive group; rows [2:9] express statistical differences by rate of increased and paired samples $t$-test.

However, after playing with Episodix, participants claimed that they would play much more with video games (40% increase for HC, 20% for MCI and 40% for AD).

In relation to motivation, before using the digital game, participants reported having a very low motivation, with the HC group obtaining the lowest score. This value rises almost 40% after the pilot experiment. In fact, in the case of the healthy control group, this figure is increased up to 50%.

On the other hand, all the participants perceived (i.e., $v = 3.5$ median value), that the use of video games for cognitive assessment seems helpful. In addition, this value is increased after the pilot experiment, especially in the control group (i.e., a 30% increase). Regarding their intention to use the digital game, since they consider it useful, they provided an average rating of 3 at the beginning, and of 4 at the end of pilot experiment.

Concerning the use of ICTs, participants indicated a median use of 3 for HC people; 3 for MCI subjects; and 1 for AD seniors. These ratings were increased by 40% after using the Episodix game.

All the parameters described were also statistically analyzed according to a paired samples $t$-test. The results are collected in Table 3 (cf. rows 6–9).

Finally, participants indicated that the Episodix game seems easier than traditional tests, in particular CVLT, and they express that they consider it more engaging and less intrusive than the pen-and-paper test (i.e., a 4 out of 5 median values).

### Preliminary psychometric study

Firstly, the initial pilot experiment allowed us to obtain relevant information to assess ecological validity, as 80% of participants found this approach closer to real life than the classical test. Similarly, subjects considered the Episodix game less intrusive than CVLT (i.e., 3.67 out of 5). Both aspects were confirmed by the healthy control group (cf. Table 3).

Secondly, with regard to criterion validity or the game's ability to predict cognitive impairments, data captured from the Episodix game was processed using statistical techniques to classify participants in healthy individuals or MCI/AD impaired subjects. First, data captured from Episodix was considered, excluding data obtained from short games played during the break between short and long-term recall. As indicated above, linear regression, random forest and support vector machines were applied (*Maroco et al., 2011*) as predictive techniques, including a study of their quality metrics (cf. Table 4).

Using a 80/20 split for training and testing (*Fan et al., 2008*), and repeating the experiment 5 and 100 times, linear regression and random forest offered best performance, especially in precision and specificity values in the case of five repetitions. For the second case, using 100 executions, results are more stable but relevant only in the random forest case (i.e., precision $= 0.74$; F1 $= 0.69$; sensitivity $= 0.65$ and specificity $= 0.82$).

On the other hand, when a training/testing ratio based on the one-left-out technique (*Soyka et al., 2012*) is applied, the most favorable results are obtained when applied to the best subset of most informative variables selected using random forest (*Archer & Kimes,*

Valladares-Rodriguez et al. (2017), *PeerJ*, DOI 10.7717/peerj.3508

**Table 4  Experiments to assess of prediction accuracy of episodix using linear regression, random forest and support vector machines.**

| | Linear regression | | | | Random forest | | | | Support vector machine | | | |
|---|---|---|---|---|---|---|---|---|---|---|---|---|
| | Precision | F1 | Sensitivity (Recall) | Specificity | Precision | F1 | Sensitivity (Recall) | Specificity | Precision | F1 | Sensitivity (Recall) | Specificity |
| Episodix [80%–20%] executions = 5 | 0.73 | 0.70 | 0.68 | 0.83 | 0.75 | 0.50 | 0.38 | 0.92 | 0.29 | 0.25 | 0.22 | 0.55 |
| Episodix [80%–20%] executions = 100 | 0.42 | 0.40 | 0.39 | 0.57 | 0.74 | 0.69 | 0.65 | 0.82 | 0.40 | 0.41 | 0.41 | 0.51 |
| Episodix [100%] | 0.58 | 0.56 | 0.54 | 0.44 | 0.69 | 0.76 | 0.85 | 0.44 | 0.55 | 0.50 | 0.46 | 0.44 |
| Episodix [best 50%] | 0.58 | 0.56 | 0.54 | 0.44 | 0.69 | 0.76 | 0.85 | 0.44 | 0.57 | 0.59 | 0.62 | 0.33 |
| Episodix (best %) LR (best %) = 5%–6% RF (best %) = 12.5% | 0.92 | 0.88 | 0.85 | 0.89 | 0.86 | 0.89 | 0.92 | 0.78 | 0.79 | 0.81 | 0.85 | 0.67 |
| Episodix +CVG [100%] | 1.00 | 1.00 | 1.00 | 1.00 | 1.00 | 1.00 | 1.00 | 1.00 | 0.90 | 0.83 | 0.77 | 0.90 |
| Episodix +CVG [best 50%] | 1.00 | 1.00 | 1.00 | 1.00 | 1.00 | 1.00 | 1.00 | 1.00 | 0.85 | 0.83 | 0.81 | 0.83 |
| Episodix +CVG (best %) LR (best %) = 50% RF (best %) = 10% | 1.00 | 1.00 | 1.00 | 1.00 | 1.00 | 1.00 | 1.00 | 1.00 | 0.90 | 0.83 | 0.77 | 0.90 |

**Notes.**
Higher quality, when accuracy values were closer to the unit.
CVG, cognitive video games during break of Episodix (similar to the break of CVLT).

*2008*). In this case, the best outcomes are obtained using linear regression using the 5%–6% of most informative variables (i.e., all accuracy metrics are higher than 0.85). However, the performance in general is very positive (i.e., all metrics above 0.85, except specificity = 0.67 for SVM).

Finally, the same technique (i.e., one-left-out) was applied on the data extracted from all games, that is, first part of Episodix, two short games to assess semantic and procedural memory, and the second part of Episodix. A data set of 89 independent samples was obtained, which enabled the correct classification of all participants independently of the subset of the most informative variables selected, using linear regression and random forest. In other words, all the quality indicators achieved their maximum value of 1.00 (cf. 6th, 7th and 8th rows in Table 4). In the case of SVM, accuracy obtained is remarkable as well, higher than 0.88 using the whole data set.

Finally, convergent validity was studied through the correlation between Episodix's variables and classical tests. As Episodix is focused on episodic memory assessment, this correlation study was focused in CVLT. As indicated above, to avoid frustration in participants with cognitive impairments, CVLT was administered to the healthy control group only. As depicted in Fig. 6, there exists a high correlation between CVLT variables and the following of Episodix variables: failures, guesses and omissions during short delay recall phases with clues; omissions from immediately recall phase during the first walk (i.e., RI-A1, RI-A2 and RI-A3) and during the second one (i.e., RI-B1); as well as time duration of short delay clued recall and free recall. Direct correlations in Fig. 6 are depicted in dark red, while indirect correlations are indicated by dark blue marks.

## DISCUSSION

This article discusses the design and initial validation of a digital game aimed to early detect MCI and AD in senior adults. For this, the game addresses the most relevant marker of early cognitive impairment, that is, episodic memory (*Albert et al., 2011*; *Hodges, 2007*; *Petersen, 2004*; *Morris, 2006*; *Facal, Guàrdia-Olmos & Juncos-Rabadán, 2015*; *Juncos-Rabadán et al., 2012*; *Campos-Magdaleno et al., 2015*). To do this, we followed an adaptation of the DS Research Methodology in IS (*Peffers et al., 2007*), which covers the whole process to obtain a working artifact both from an ICT and a medical perspective. For instance, regarding the initial methodological step to define solutions, gamification techniques were introduced due to the limitations of classical tests. In addition, a user-centered design methodology was applied and usability was considered as a design priority due to the target users' profile (*Costa et al., 2017*). Furthermore, being aware of the lack of psychometric maturity (*Valladares-Rodríguez et al., 2016*) of the case studies found in the literature (*Plancher et al., 2012*; *Plancher et al., 2013*; *Sauzéon et al., 2015*; *Jebara et al., 2014*), this aspect was instrumental to guide both the design and implementation processes.
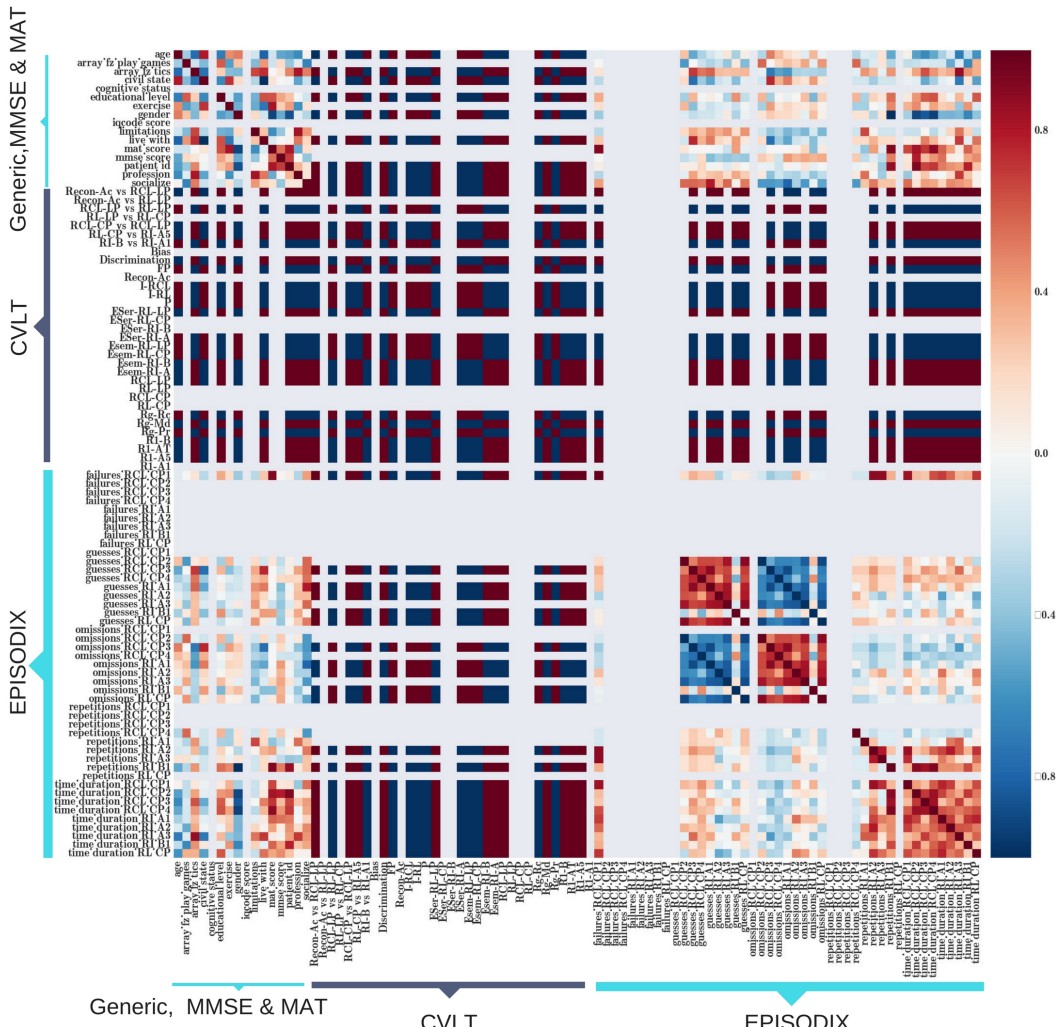

**Figure 6  Heat map of correlation between Episodix's variables and classical tests: CVLT, MMSE and MAT.**

The game corresponds to the gamification of one of the most popular tests for the assessment of episodic memory (i.e., CVLT). Some key aspects were modified to obtain a more ecological tool and to better reflect the complexity of this type of memory. For example, instead of verbally reproducing an imaginary shopping list, scheduled for Monday and Tuesday, the subject being tested is invited to take a virtual walk where objects to be eventually recalled are naturally present on the street or in stores, that is, mimicking real life situations. Furthermore, categories and objects were updated so that they are easier to recognize visually, but always maintaining the same frequency of use as in the original test. For example, categories of vehicles and stores are included, instead of herbs and tools. Incidentally, the strategy implemented to dynamically generate object lists eliminates the practice effect in successive administrations. Note that object

lists in Episodix are not influenced by confounding factors, and more specifically age and educational level, as the categories selected are neutral with respect to these two factors. Finally, the administration time is dramatically reduced with respect to CLVT's, with game-based testing sessions lasting from 30 to a maximum of 45 min. Note that, considering the additional post-processing time that pen-and-paper tests require, total administration + diagnosis time is dramatically cut to less than a half the original time. In addition, Episodix game was perceived as a friendly application by our target users, that is, senior adults. As a consequence, the intrusive characteristics of classical pen-and-paper tests is avoided. Thus, the whole features above contribute to guarantee content validity and usability of this approach, overcoming at the same time the limitations of the original pen-and-paper test.

Regarding the design methodology, the two focus group sessions carried out provided key information to validate and drive the design and implementation of Episodix. Participants were enrolled from two different sources, namely a university elders' program and a senior association. As a consequence, a representative sample of the expectations and needs of senior adults was considered. From a psychometric point of view, the approach was validated by assessing content validity, where practically all participants enjoyed the idea of taking a virtual walk, the new semantic categories, and the new approach being a visual tool rather than a pen and paper test; face validity, where almost all participants perceived the digital game as an useful tool; and finally ecological validity, when all participants considered both the digital tool and the new objects close to real life, and also less intrusive than conventional tests. With regard to usability, almost all participants found Episodix easily administrable, even to feel capable of interacting with it without assistance. Note that focus group participants had a negative initial perception about information technologies in general, and video games in particular. This perception dramatically improved when they were introduced to the game and they realized its cognitive utility. They also expressed their fears to obtain a negative assessment or being alerted about a cognitive impairment. In other words, the game is perceived as very useful, but they would prefer it to be administered by a health professional better than themselves, no matter they would be able to interact with it on their own. In sum, focus groups helped to improve the usability and psychometric validity of the game, but they also confirmed an existing digital divide among senior adults.

To validate the first functional prototype of Episodix, a pilot experiment was carried with a cohort sample involving 16 people. Despite its limited size, the sample fulfills the basic requirements on variability and heterogeneity with respect to age, gender, educational level or cognitive status. The initial interview and the administration of classical tests confirmed that cognitive impairment is correlated to age, and that usage frequency of new technologies decreases as cognitive impairment increases; the same occurs in the frequency of use of games, except for the MCI group. Finally, it was also confirmed that the external variables related to the incidence of cognitive impairment considered (i.e., educational level, socialization level and physical exercise level), were indeed (inversely) correlated.

The pilot experiment also included a comparison of the perceived usability prior and after interacting with the game (*Lee, Kozar & Larsen, 2003*). The motivation and the intention to interact with games and the perceived usefulness of games dramatically increased after interacting with Episodix. The final satisfaction survey confirmed that Episodix was perceived as more ecological, engaging, easier and less intrusive than the original pen-and-paper test, with individual median values ranging from 3.5 to 4 in a 1 to 5 scale. In parallel, the significance of the increase between pre- and post-pilot perceptions was measured by means of the statistical paired samples $t$-test. The main outcomes show that this increase is significant in willingness to play more video games (i.e., $t = 0.000426$); motivation to use video games (i.e., $t = 0.001291$); and finally, perception of considering this kind of cognitive games as useful tools (i.e., $t = 0.040568$).

The experiment also provided information on ecological validity from end users' perspective. In this regard, the results were highly favorable since all participants found this approach closer to real life than the classical test, with a median value of 4 out of 5. Thus, it can be considered an ecologic tool from seniors point of view.

Furthermore, data captured during the pilot—including variables captured from game interactions and non-cognitive external variables—served to assess the predictive and classification capabilities of Episodix, that is, its capability to detect cognitive impairments. The ability to correctly classify individual subjects was highly satisfactory, especially when the one-left-out training/testing methodology was applied. Using the best percentage of the most informative variables, rising considerably all quality metrics of algorithms calculated. It is especially striking the increase in almost 4 points the quality, when we use the best % instead the 50% of informative data for the three algorithms: LR (i.e., 5%–6%); RF (i.e., 12.5%) and SVM (i.e., 12%). Concurrently, if the variables obtained from the interaction with the two short games administered during the Episodix break were also considered, precision, specificity and sensitivity reach the maximum value of 1.00 for LR and RF, reducing a little for SVM, but all quality parameters are higher than 0.88. In other words, the suite composed of the three games correctly classified all participants. Besides, 2 people were correctly diagnosed with MCI using exclusively game data, which was confirmed by administering classical tests. Again, the pilot experiment confirmed the acceptability of Episodix and provided a preliminary psychometric validation about criterion and ecological issues.

Finally, data about the convergent validity of Episodix was also obtained. The best correlations between CVLT and Episodix variables were, on the one hand, failures, guesses and omissions during short delay recall phases with clues, and on the other hand, omissions from immediate recall phase during all trials of first and second walks. Moreover, time duration provides a good correlation with CVLT, particularly during short delay clued phase and free recall phase. Note that CVLT is broadly adopted as the golden standard.

## LIMITATIONS

No matter the highly promising results discussed above, some issues were detected during the design and validation process that shall be further investigated. Firstly, with respect to accessibility, senior adults clearly prefer a touch interface better than traditional keyboard and mouse. Participants both from focus groups and pilot experiment indicated tan a touch interface would avoid the appearance of a confounding factor related to the technological level of the subject.

Other aspect to be considered is the limited size of the sample. At the time of writing this paper we are in the process of designing and implementing a more representative pilot experiment, targeted to identify a broader range of cognitive impairments in participating subjects, as well as a comprehensive validation of the psychometric properties of the digital game.

Finally, some improvements on the presentation speed of stimuli, the description of the tasks to be performed in the virtual world, and on some aspects of the game mechanics are being introduced to further facilitate self-administration and, as a consequence, to further facilitate the administration by health professionals.

## CONCLUSIONS

The initial evidence obtained enables us to conclude that the game developed and implemented as an episodic memory evaluation tool may suppose a relevant and innovative breakthrough, as it overcomes the main limitations of paper-and-pencil tests and other previous approaches based on gamification discussed in this article, and dramatically reduces administration and diagnosis time requirements. Furthermore, exploratory results are also promising from the perspective of an early prediction of MCI and AD. The research questions posed at the beginning of this research has been responded. On the one side, a gamified, improved version of CVLT was implemented to assess episodic memory in a way similar to the original pen-and-paper method. On the other side, a first pilot experiment correctly classified all participating subject according to their cognitive status. The new approach is time-effective, ecological, non-intrusive, overcomes practice effect and is not affected by most common confounding factors.

Presently we are investigating the application of additional machine learning techniques to perform cognitive classification or assessment from the variables obtained from the game, apart from the ones introduced in this article. We are also developing a new pilot experiment to obtain sufficient evidence to guarantee clinical-grade psychometric and normative validity of the approach discussed in this paper.

## ACKNOWLEDGEMENTS

Authors appreciate the support of AFAGA (the Association of Family Members of Alzheimer Patients and other Dementias of Galicia, Spain), which provided voluntary participants among its members. We will also like to thank Mr. Luis Pereiro Melon for

his contribution to the software development of digital games, and Dr. Carlos Rivas Costa for his encouragement and support during the pilot experiment. Finally, we will like to thank the valuable comments and suggestions provided by experts during the design phase of digital game: Dr. Arturo X. Pereiro Rozas and Dr. Cristina Lojo Seoane from the Department of Developmental Psychology at the University of Santiago de Compostela, and MDM Jose Moreno Carretero from the Neurology service at the Alvaro Cunqueiro University Hospital (Vigo, Galicia, Spain).

The group of the Signal and Communications Theory (GTM: http://gtm_voz.webs. uvigo.es/) from the School of Telecommunication Engineering at the University of Vigo provided invaluable assistance to obtain text transcriptions from the audio files collecting the focus group sessions.

# APPENDIX A. STREET MAP OF EPISODIX' SMALL CITY

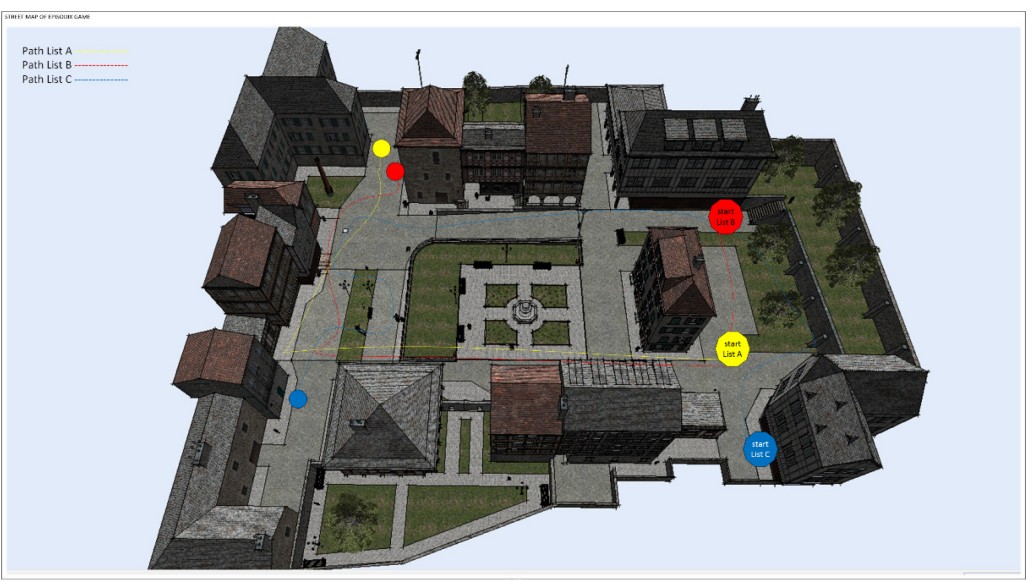

**Figure A1**  **Street map of path lists in the Episodix digital game.**

# APPENDIX B. LIST OF VARIABLES CAPTURED IN THE EPISODIX GAME

**Table A1** **List of variables captured during the administration of the Episodix digital game.**

**EPISODIX**

**Variables captured from interacting with game ordered by most informative**

| | |
|---|---|
| 1. time_duration_RI_B1 | 24. omissions_RCL_CP1 |
| 2. time_duration_RCL_CP2 | 25. guesses_RI_A3 |
| 3. repetitions_RL_CP | 26. guesses_RI_A2 |
| 4. time_duration_RCL_CP4 | 27. omissions_RI_A2 |
| 5. repetitions_RI_B1 | 28. omissions_RI_A3 |
| 6. time_duration_RCL_CP1 | 29. omissions_RCL_CP3 |
| 7. time_duration_RI_A2 | 30. guesses_RCL_CP3 |
| 8. time_duration_RCL_CP3 | 31. repetitions_RI_A1 |
| 9. time_duration_RI_A3 | 32. failures_RCL_CP4 |
| 10. array_fz_play_games | 33. repetitions_RI_A2 |
| 11. time_duration_RL_CP | 34. repetitions_RI_A3 |
| 12. time_duration_RI_A1 | 35. guesses_RCL_CP4 |
| 13. omissions_RCL_CP2 | 36. omissions_RCL_CP4 |
| 14. guesses_RCL_CP2 | 37. failures_RCL_CP2 |
| 15. omissions_RL_CP | 38. repetitions_RCL_CP4 |
| 16. guesses_RI_B1 | 39. repetitions_RCL_CP2 |
| 17. omissions_RI_B1 | 40. repetitions_RCL_CP1 |
| 18. omissions_RI_A1 | 41. repetitions_RCL_CP3 |
| 19. guesses_RL_CP | 42. failures_RI_A1 |
| 20. guesses_RI_A1 | 43. failures_RI_A2 |
| 21. failures_RCL_CP3 | 44. failures_RI_A3 |
| 22. failures_RCL_CP1 | 45. failures_RI_B1 |
| 23. guesses_RCL_CP1 | 46. failures_RL_CP |

**Comparative of variables generated automatically from interacting with game and correlative ones with CVLT**

| EPISODIX | CVLT (*Woods et al., 2006*) |
|---|---|
| Total trials 1–3 | Total trials 1–5 |
| Short-delayed free recall | Short-delayed free recall |
| Short-delayed cued recall | Short-delayed cued recall |
| Long-delayed free recall | Long-delayed free recall |
| Long-delayed cued recall | Long-delayed cued recall |
| Total recognition discrimination | Total recognition discrimination |
| Total repetitions | Total repetitions |

# APPENDIX C. NEW DESIGNED WORD LISTS FOR THE EPISODIX GAME

## List A and List B

The following list shows the comparison between List A and List B from the original test and Episodix. Both have a similar average frequency of use. It was calculated thorough Google books Ngram Viewer: https://books.google.com/ngrams.

**Table A2   Comparative List A and List B from CVLT vs Episodix game.**

|  | CVLT-A | av. fr. | EPISODIX-A | av. fr. | CVLT-B | av. fr. | EPISODIX-B | av. fr. |
|---|---|---|---|---|---|---|---|---|
| 1 | Drill | 1.4E–06 | bus | 7.7E–05 | Skimmer | 1.4E–05 | Sidecar | 1.9E–06 |
| 2 | Lemons | 1.1E–04 | Musician | 4.2E–04 | Cherries | 6.6E–05 | Kindergarten | 4.0E–05 |
| 3 | Jacket | 2.8E–05 | Fountain | 4.3E–03 | Tuna | 5.2E–05 | Roundabout | 3.0E–01 |
| 4 | Saffron | 5.5E–05 | Restaurant | 1.6E–04 | Mint | 7.6E–06 | Paintbrush | 4.8E–05 |
| 5 | Grapes | 3.1E–04 | postman (male) | 6.1E–05 | kiwis | 1.0E–06 | Police station | 1.4E–04 |
| 6 | Cumin | 4.1E–05 | market | 2.7E–05 | Mixer | 7.6E–06 | Skates | 1.9E–05 |
| 7 | Socks | 1.5E–03 | stop | 4.4E–04 | Garlic | 1.6E–04 | Pulley | 1.0E–04 |
| 8 | Shovel | 3.2E–04 | bicycle | 8.9E–05 | Flounder | 9.4E–06 | Sewer | 5.9E–01 |
| 9 | Laurel | 4.3E–04 | Bakery | 8.0E–05 | Paprika | 1.8E–05 | Hose | 3.1E–02 |
| 10 | Mandarins | 1.3E–05 | builder (male) | 1.2E–04 | Strawberry | 5.9E–05 | chimney | 3.2E–04 |
| 11 | Saw | 1.8E–03 | motorbike | 4.5E–05 | Megrim | 2.3E–04 | Pot | 1.6E–01 |
| 12 | Shoes | 6.1E–04 | Bin | 3.7E–05 | Dishes | 3.9E–04 | Wheelbarrow | 9.2E–07 |
| 13 | Rosemary | 1.0E–04 | Greengrocer | 4.6E–06 | Apricots | 2.3E–05 | Auditorium | 5.1E–04 |
| 14 | Pineapple | 8.3E–05 | fireman (male) | 2.3E–05 | Trout | 4.6E–05 | Mailbox | 2.8E–01 |
| 15 | Screws | 1.6E–04 | Skateboard | 7.2E–07 | oregano | 3.1E–05 | Cement-mixer | 1.6E–06 |
| 16 | Gloves | 2.2E–04 | Lamps | 1.5E–05 | Casserole | 7.9E–05 | Tricycle | 6.8E–06 |
|  | fr media | 3.6E–04 | fr media | 3.7E–04 | fr media | 7.5E–05 | fr media | 8.5E–05 |

## List C/recognition list

The list below shows the comparison between List C from the original test and Episodix. As in the previous case, average frequencies are comparable.

**Table A3  Comparative List C from CVLT vs Episodix game.**

| | CVLT-C | Type | av. fr. | EPISODIX-C | Type | av. fr. |
|---|---|---|---|---|---|---|
| 1 | Shoes | A | 6.1E–04 | Bin | A | 3.7E–01 |
| 2 | Oregano | BC | 3.1E–05 | Sidecar | BC | 1.9E–02 |
| 3 | Megrim | NC | 9.4E–06 | Kindergarten | NC | 4.0E–01 |
| 4 | Clock | NR | 7.0E–04 | Clock | NR | 7.0E–04 |
| 5 | Land | RF | 3.1E–02 | Hose | RF | 4.6E–05 |
| 6 | Cinnamon | PR | 2.7E–04 | Ice-cream-shop | PR | 3.0E–06 |
| 7 | Socks | A | 1.5E–03 | Stop | A | 4.4E–04 |
| 8 | Sheets | NR | 1.9E–04 | Roses | NR | 6.8E–04 |
| 9 | rocking chair | RF | 1.9E–05 | Bridge | RF | 3.0E–03 |
| 10 | Shovel | A | 3.2E–04 | Bicycle | A | 8.9E–05 |
| 11 | Mandarins | A | 1.3E–05 | Builder (male) | A | 1.2E–04 |
| 12 | Strawberry | NC | 5.9E–05 | Pulley | NC | 1.0E–04 |
| 13 | Casserole | BC | 7.9E–05 | Sewer | BC | 5.9E–01 |
| 14 | Bonbons | RF | 2.4E–05 | Waiter (male) | RF | 1.2E–04 |
| 15 | Cumin | A | 4.1E–05 | Market | A | 2.7E–02 |
| 16 | Books | NR | 1.1E–02 | Books | NR | 1.1E–02 |
| 17 | Drill | A | 1.4E–06 | Bus | A | 7.7E–05 |
| 18 | Vitamins | NR | 1.7E–04 | Stairs | NR | 3.5E–04 |
| 19 | Carnation | RF | 9.0E–05 | Drums | RF | 7.0E–04 |
| 20 | Grapes | A | 3.1E–04 | Postman (male) | A | 6.1E–01 |
| 21 | Thread | NR | 1.7E–03 | Glasses | NR | 8.2E–05 |
| 22 | Blazer | PR | 1.6E–04 | Cone | PR | 6.5E–04 |
| 23 | lemons | A | 1.1E–04 | Musician | A | 4.2E–04 |
| 24 | Trout | NC | 4.6E–05 | Chimney | NC | 3.2E–04 |
| 25 | Saffron | A | 5.5E–05 | Restaurant | A | 1.6E–04 |
| 26 | Whistles | RF | 1.8E–05 | Paper | RF | 1.2E–02 |
| 27 | Garlic | BC | 1.6E–04 | Wheelbarrow | BC | 9.2E–03 |
| 28 | Jacket | A | 2.8E–05 | Fountain | A | 4.3E–03 |
| 29 | Carpet | NR | 2.0E–04 | Carpet | NR | 2.0E–04 |
| 30 | Rosemary | A | 1.0E–04 | Greengrocer | A | 4.6E–02 |
| 31 | Gloves | A | 2.2E–04 | Lamps | A | 1.5E–02 |
| 32 | Apples | PR | 4.4E–04 | Teacher (female) | PR | 2.6E–04 |
| 33 | Sticks | RF | 3.8E–05 | Jacket | RF | 1.6E–04 |
| 34 | Pineapple | A | 8.3E–05 | Fireman (male) | A | 2.3E–01 |
| 35 | Rosemary | A | 1.8E–03 | Motorbike | A | 4.5E–05 |
| 36 | Apricots | BC | 2.3E–05 | Mailbox | BC | 2.8E–01 |
| 37 | Aspirins | RF | 5.9E–06 | Bulb | RF | 3.0E–05 |

**Table A3** (*continued*)

|   | CVLT-C | Type | av. fr. | EPISODIX-C | Type | av. fr. |
|---|--------|------|---------|------------|------|---------|
| 38 | Wallet | NR | 6.5E–04 | Wallet | NR | 6.5E–04 |
| 39 | Screws | A | 1.6E–04 | Skateboard | A | 7.2E–07 |
| 40 | Mixer | NC | 7.6E–06 | Cement-mixer | NC | 1.6E–02 |
| 41 | Tongs | PR | 1.0E–04 | Lorry | PR | 2.3E–04 |
| 42 | Laurel | A | 4.3E–04 | Bakery | A | 8.0E–02 |
| 43 | Duster | RF | 2.3E–05 | Bookshop | RF | 4.3E–04 |
| 44 | Soap | NR | 3.1E–04 | Cardboard | NR | 2.7E–04 |
|   |   | **av. fr.** | **1.2E–03** |   | **av. fr.** | **8.6E–04** |

**Notes.**

A, word from list A; BC, word from list B, belonging to common categories; NC, word from list B, belonging to non-common categories; PR, prototype word; RF, word with a phonetic relationship; NR, non-relation word.

# APPENDIX D: SCRIPT DEVELOPMENT FOR FOCUS GROUP SESSIONS

**Table A4** **Script of the focus group sessions performed.**

**Focus group script**

**Presentation**

Introduction of the moderator and description of the work/research being carried out.
Introduction of participants: name, age, occupation, hobbies, etc.

**Opinions on video games and ITC**.

To start with, I would like to talk about new technologies or ITC and video games:
Do you like new technologies? Do you think that new technologies mya improve or positively influence your quality if life? How do you manage with a computer, a mobile phone or a tablet computer?
For what may video games could be useful? Which video games do you usually play? Which video games would you like to play if possible? What are your motivations to play video games? Do you know virtual reality video games that could make us feel that we are inside the game?

**Opinions on memory assessment**

Next I would like to talk about people skills such us memory, attention, language, etc,
and how such abilities could be studied. I would like to know your opinion about:
Are you interested in how memory capabilities are evaluated? Why? Are you worried about the detection of some memory or attention problem? Which activities do you perform to keep your brain active?

**Comparison between classical vs. video-game-based evaluation**

I would like to propose a simple exercise. For this, I will first briefly describe how episodic memory is evaluated in a clinical setting. Then, I will show you a preliminary version of a video game with the same objective.

I show images from TAVEC and introduce the game mechanics.
Then, I show a first execution of the Episodic prototype.

**Table A4** (*continued*)

| Focus group script |
| --- |
| **Final opinion on the practical experience** |
| Now I would like to talk about your opinions on the game I have just shown to you, and about your experiences with this game. |
| What do you think about cognitive questionnaires or tests such as TAVEC? Do they recall to you some daily life activity? Which? How would you improve these tests? |
| What do you think about ''Episodix''? Do you like the scenery? Which improvements or changes would you propose? Do it recall to you some daily life activity? Which? Do you feel it is invasive or alien? |
| About using it. DO you think it is intuitive? How would you like to play this game? Using a Mobile phone, a tablet, a TV, a PC, another device? |
| Would you like to use a collection of video games to assess your memory, attention, etc.? Do you think that Episodix serves to evaluate memory? |
| Finally, do you wish that researchers keep investigating this topic? Would you like to participate in a future validation of this type of systems? |
| **I warmly thank you for your collaboration. You will receive a summary of the results of this session** |

## Funding

The present work has been funded by the government of Galicia (Xunta de Galicia, Spain) to cover the travel expenses to participants' homes during the pilot experiment (Ref.: 2016/236). The funders had no role in study design, data collection and analysis, decision to publish, or preparation of the manuscript.

## Grant Disclosures

The following grant information was disclosed by the authors:
government of Galicia (Xunta de Galicia, Spain): 2016/236.

## Competing Interests

The authors declare there are no competing interests.

## Author Contributions

- Sonia Valladares-Rodriguez conceived and designed the experiments, performed the experiments, analyzed the data, contributed reagents/materials/analysis tools, wrote the paper, prepared figures and/or tables, reviewed drafts of the paper, contacts with eldery associations.
- Roberto Perez-Rodriguez conceived and designed the experiments, analyzed the data.
- David Facal conceived and designed the experiments, analyzed the data, contributed reagents/materials/analysis tools, wrote the paper, reviewed drafts of the paper.
- Manuel J. Fernandez-Iglesias analyzed the data, wrote the paper, reviewed drafts of the paper.
- Luis Anido-Rifon analyzed the data, wrote the paper, reviewed drafts of the paper, acquiring funds for research.
- Marcos Mouriño-Garcia performed the experiments, analyzed the data, contributed reagents/materials/analysis tools.

## Human Ethics

The following information was supplied relating to ethical approvals (i.e., approving body and any reference numbers):

The University of Vigo granted Ethical approval to carry out the study (Galician Ethics Committee (Spain) Ref: 2016/236).

## Data Availability

The raw data has been uploaded as a Supplementary File.

## Supplemental Information

Supplemental information for this article can be found online at http://dx.doi.org/10.7717/peerj.3508#supplemental-information.

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
