# Peer review of "Design process and preliminary psychometric study of a video game to detect cognitive impairment in senior adults"

_PeerJ, doi:10.7717/peerj.3508_

## Round 0.1 · original submission · Major Revisions

Dear Authors,

Two peer reviewers have given comments that are very important to improve your manuscript in the methodology, data analysis and discussion sections.

Please read their reviews and proceed to the major revisions needed of your manuscript.

·

Basic reporting

The English language in some sentences should be improved to ensure international audience can clearly understand the text, including lines 66 (...novel research line...) and 500 (No matter the highly promising results discussed above,...).

There is a lack of literature and references in the paper which are associated with the use of games in measuring cognitive impairments, or at least in neuroscience.

The authors tried to mimic the testing mechanics of CVLT when designing the game, without explaining how they made the game fun and challenging (line 97). The explanation is important to justify that Episodix is actually a video game, as opposed to a digital test (see line 295).

Experimental design

The title of the paper is misleading, as the title suggested a study to examine the effect of a game in predicting cognitive impairments, but only Section 3.3.3 is actually written about an initial psychometric study with 89 samples, while the rest of the paper were discussing the focus group conducted with 16 subjects to gather opinion for validating the game.

If the paper intends to focus on the prediction of cognitive impairments, this section should be expanded to include how the scores in the game world were benchmarked with the results of a pen-and-paper test of CVLT. As an evidence of gamification, the alignment of structural elements of games, i.e. game goal, rules, challenge, feedback and interaction to the CVLT should be explained.

Validity of the findings

The goal of the focus group was to gather opinions on the use of video games in cognitive evaluation. The authors should streamline the paper on this splitting as opposed to comparing the effects between paper-and-pen test and digital test in predicting cognitive impairments.

The contents used in the game should not be validated by the subjects since they generally do not possess the expert knowledge needed to decide what should be included in the game or test; contents should be validated by authorised experts in the CVLT.

Additional comments

This paper contains good materials for a journal, but it consists of multiple aims, which diverted readers' attention to a certain extent. The author may consider splitting the paper into two, one concentrates on the design, development and validation of Episodix, while another focuses on the usability tests on the game, conducted with 16 subjects. However, in either split of the paper, the literature review should be reinforced by referring to best practices on gamification in healthcare.

·

Basic reporting

The overall structure of the paper is sound and provides a good framework for reporting the reasoning behind the study, the details of the experimental protocol, and the findings and conclusions drawn from them.

The place where I think this paper could benefit the most from further work is with regard to the language in general, grammar in particular, and the figures that are included in the paper.

I think it would be very helpful to the reader if the paper were subjected to a rigorous review in terms of the English language employed. This would increase the readability of the paper and help avoid confusion for the readers, in terms of the design and findings of the described study. In the current version, I would sometimes be unsure about what the authors meant, or I would be distracted by the language employed to the extent that I lost focus while going through the paper.

Additionally, I would suggest considering whether all the included content is strictly necessary. For instance, section 2.3 goes to great lengths to provide text-book definitions of reliability and validity. I would expect most readers of the paper to be satisfied with more succinct descriptions. I do appreciate the content on lines 213-232, where the considerations on validity and reliability become specific to the game in question, though.

I would also suggest considering whether Fig. 1 and Fig. 3 are helpful to the reader. For most studies of this kind, I usually consider the reasoning behind the experiment, the experimental design, the protocol employed, and the findings of the study to be most relevant and interesting content. Software architecture is usually less relevant in that most design decisions are either drawn from well-known templates or feature implementations that are wholly specific to the project in question and thus less transferable to other work. For this particular paper, which does treat a very interesting idea and study, I would recommend reducing the details of the software architecture to the bare minimum. I feel this would help the reader focus on the important parts of the paper, namely the hypotheses motivating the work, the designed game and the experimental design, and the obtained results.

While on the topic of figures, I think that more descriptive figure captions could be a change that would be useful to the reader. E.g. Fig. 8 is captioned with the text "Different metrics about the prediction accuracy of the Episodix game".
Here, I think it would be helpful if the caption offered some information to the four metrics displayed and what can be read from the results in the figure.
Also, both axes in this particular figure are missing labels, and they need to be labeled to avoid confusion.

In this same results section, it is hard for me to understand from "3.3.3. Initial psychometric study" exactly what is being predicted.

Moving from the topic of figures to the topic of tables, I think the size of the tables could be reduced significantly be removing some of the decimals from the findings as these don't really communicate much information.

Lastly, I would prefer a cutback on the technical description of the game (as noted above) and greater detail on the particular design of the Episodix game. I would like to learn more about the first person experience of interacting with the game, as this seems much more relevant for the reader when they are trying to make sense of how the approach is different from a typical pen-and-paper test.

Altogether, I think the reporting of the paper could benefit from a thorough iteration, but that being said the fundamental structure is sound.

Experimental design

The experimental design looks suitable for the question that the paper is investigating.

I appreciate that the study is trying to approach the problem of using Episodix holistically, investigating the properties of the design, using participatory design, investigating the attitudes of the participants before and after the study, and investigating whether a regression model can be built to predict the condition of the participants.

I would say, however, that the reporting makes it hard to completely evaluate the findings. It is not communicated very clearly what is being predicted, but it is my understanding that you are attempting to classify the participants using a regression model. While this is possible, it was not completely clear to me what motivated this choice. When you report the precision, recall, F1, etc. it's not clear to me what they are calculated relative too. I think you need to do some work to communicate this more clearly to the reader. Additionally, when you tell me that you are communicating the "most outstanding results", I'm wondering about the cases where your results were less outstanding. Please make sure to report all your findings in a comprehensive way, and also provide reports of the cases where your method failed or had a hard time making acceptable predictions. This could go in section 5, but I also think it should be addressed in the results section.

Validity of the findings

It is hard for me to evaluate the validity of the findings with the current style of reporting. I think it would be helpful to revisit the prose of the paper, improve tables and figures, provide a more clear description of the predictive parts of the work. With these improvements, I would be able to provide a better comment on the validity of the findings.

I will say, though, that the overall design of the study seems promising, as I understand it, so I see the main challenges being with regard to reporting and analysis.

Additional comments

Altogether I think this is a paper with a fundamentally interesting, challenging, relevant, and appealing idea. Having better/different methods for characterizing cognitive decline could be highly helpful, especially as many societies are seeing a move toward a larger part of their population being elderly.
The holistic approach to the problem as well as the concern with validity and reliability are very appealing to me, and it also seems that the data collection was fundamentally sound.

What I think would improve this paper would be to reduce the reporting of details that I would categorize as software engineering, improve the reporting in general to make it more salient exactly what the hypotheses of the study were and what methods were employed, and improve the reporting and analysis in terms of results. Possibly also choose a different method of analysis than linear regression, there are many more powerful tools out there that seem like they would be more appropriate for you problem as I understand it.

Please don't be disheartened by this slightly negative review, I think there is a valuable study here that could be interesting to the community if some work is put into reporting and analysis.

I am disclosing my name with this review, please feel free to reach out if you would like to discuss the paper in further detail.

---

## Round 0.2 · Minor Revisions

Dear Authors,

Enclosed are suggestions to further improve your manuscript.Please do so and resubmit the revised manuscript as soon it is done.

·

Basic reporting

The revision greatly improved the readability of the paper. Relevant literature references were added to increase its connection to both the field of gamification and the field of its content knowledge. The structure, figs, tables and raw data were made available and checked, indicating its readiness for professional academic publication. No hypothesis was tested in this article.

Experimental design

Since the authors decided not to streamline the article to focus on specific methods, there should be a priority set for the combination of pilot experiment, focus group and usability test. Which was the dominant research method in this study?

Validity of the findings

There should be an explanation of how results of each method complement or triangulate results generated from other methods, forming a coherent set of findings that can answer the research question: Can a videogame be designed and developed to assess episodic memory to predict early cognitive impairments in an ecological and non-intrusive way?

The Conclusion section actually indicated the answer of the research question, but the linkage to the findings needs to be strengthened, probably by resolving the issues mentioned in 2. Experimental Design and 3. Validity of the findings.

Additional comments

This version of the article shows great improvement as compared to the previous version.

---

## Round 0.3 · accepted · Accept

Dear Authors,

Thank you for the revisions made and I am happy to inform that the manuscript has been accepted for publication in PeerJ.

Thanking You

·

Basic reporting

No comment.

Experimental design

Accepted the revision made.

Validity of the findings

Accepted the revision made.

Additional comments

No comment.